# Phenolic Resin Foam Composites Reinforced by Acetylated Poplar Fiber with High Mechanical Properties, Low Pulverization Ratio, and Good Thermal Insulation and Flame Retardant Performance

**DOI:** 10.3390/ma13010148

**Published:** 2019-12-31

**Authors:** Jian Liu, Liuliu Wang, Wei Zhang, Yanming Han

**Affiliations:** 1Ministry of Education Key Laboratory of Wood Materials Science and Utilization, Beijing Forestry University, Beijing 100083, China; liujian@bjfu.edu.cn (J.L.); wangliuliu18@126.com (L.W.); 2Beijing Key Laboratory of Wood Science and Engineering, Beijing Forestry University, Beijing 100083, China; 3Research Institute of Forestry New Technology, Chinese Academy of Forestry, Xiangshan Road, Beijing 100091, China

**Keywords:** poplar fiber, surface compatibilization, phenolic resin foam, toughening

## Abstract

Phenolic foam composites (PFs) are of substantial interest due to their uniform closed-cell structure, low thermal conductivity, and good thermal insulation performance. However, their disadvantages of a high pulverization rate and poor mechanical properties restrict their application in building exterior insulation. Therefore, the toughening of these composites is necessary. In this paper, poplar fiber was treated with an acetylation reagent, and the acetylated fiber was used to prepare modified phenolic foams (FTPFs); this successfully solved the phenomenon of the destruction of the foam structure due to the agglomeration of poplar fiber in the resin substrate. The foam composites were comprehensively evaluated via the characterization of their chemical structures, surface morphologies, mechanical properties, thermal conductivities, and flame retardant properties. It was found that the compressive strength and compressive modulus of FTPF-5% respectively increased by 28.5% and 37.9% as compared with those of PF. The pulverization ratio was reduced by 32.3%, and the thermal insulation performance and flame retardant performance (LOI) were improved. Compared with other toughening methods for phenolic foam composites, the phenolic foam composites modified with surface-compatibilized poplar fiber offer a novel strategy for the value-added utilization of woody fiber, and improve the toughness and industrial viability of phenolic foam.

## 1. Introduction

Energy resources are the material basis of human survival and development, and the key to sustainable economic development. All the countries of the world are actively responding to the call of energy conservation and emission reduction [1]. At present, building energy consumption is substantial, accounting for 20.6% of China’s total energy consumption (899 million tons of standard coal), and it is expected that the proportion of building energy consumption in the total national energy consumption will rise to 40% in 2020. According to this situation, there is an urgent need for building energy conservation [2]. As an important component of energy saving and emission reduction in the field of construction, exterior wall insulation has attracted extensive attention and provides effective solutions. Organic thermal insulation composites with low thermal conductivity and a good thermal insulation effect have been significantly developed. Styrene foam (PS) and polyurethane foam (PU) have gradually become the most widely used building thermal insulation composites. However, due to their poor heat resistance, easy combustion, and the release of large amounts of smoke and toxic gas after combustion, these two composites easily result in huge losses [3].

Phenolic foam (PF), an excellent fire insulation composite, has reached the standard of Class B1 flame-retardant foam without modification [4]. It has a uniform closed pore structure, low thermal conductivity, and good thermal insulation performance, and is therefore better than PS and PU. Even in the event of a fire, phenolic foam shows no dripping and less smoking, and the char layer formed on its surface can protect the internal structure [5,6]. Phenolic foam is a thermal insulation composite with great development potential; it is known as a third-generation emerging thermal insulation composite, and is widely used in various thermal insulation pipelines [7], transportation [8], and flowers [9]. However, it is difficult to use phenolic foam in the direct preparation of thermal insulation exterior wall materials due to the defects of the material itself; it exhibits poor mechanical properties, high brittleness, and a high pulverization rate, which seriously affect its application [10]. Therefore, reinforcement has been an important research direction of phenolic foam composites.

In recent years, many researchers have conducted numerous studies on the strengthening of phenolic foam composites. The easiest and most convenient way to enhance phenolic foam is to blend some external toughening agents in phenolic resins, such as natural rubber, nitrile rubber [11,12,13,14], external flexures, and fiber materials, such as glass fiber [15,16,17]. Although the addition of rubber improves the toughness of PF, it reduces its heat resistance [18]. Similarly, while the addition of short glass fibers enhances the toughness and mechanical strength of PF, it reduces its thermal insulation properties. The addition of short fibers as a reinforcing material for phenolic foam can reduce its brittleness and improve its shear strength [19]. Natural fiber has the advantages of light weight, low cost, and reproducibility. With cellulose microfiber as a reinforcer, and lignin and hemicellulose as the matrix, the resulting composite has the characteristics of a high specific strength and high specific modulus. Therefore, it is an excellent environmentally friendly reinforcing material. At present, natural fibers (such as sisal fiber, jute fiber, ramie fiber, and straw fiber, among others) have been used more often to strengthen rigid polyurethane foam [20,21,22,23,24]. The resulting modified polyurethane foam has obvious enhancements, is non-toxic and environmentally friendly, and the degradation rate of composite materials is obviously improved. However, as plant fibers contain a large number of hydroxyl groups and have a relatively high polarity, they can easily aggregate as a filling material, and complexing with weak polar organic polymers, such as resin, is relatively difficult. Therefore, it is necessary to modify the interface of plant fiber via pretreatment to enhance the interfacial compatibility between fibers and resin. The pretreatment method of acetylation modification has the advantages of a simple reaction, a short reaction time, a good modification effect, no toxicity, and little pollution [25,26,27,28,29]. Acetic anhydride has been used to react with the hydroxyl groups on the surface of fiber, and acetyl groups are grafted onto the surface to form an esterification layer to reduce the polarity of the fiber. This is conducive to the better fusion of fibers between resins, and could avoid the large holes between foam caused by fiber agglomeration. The composite material prepared by acetylated plant fiber and polyester resin matrix is significantly stronger than that without modified fibers. Therefore, acetylated modified plant fibers are expected to improve the mechanical properties and reduce the fragility of phenolic foam.

In this study, phenolic foams with different contents of fiber and acetylated fiber were prepared. The chemical structures, morphological characteristics, mechanical properties, and thermal properties of PF and wood fiber-reinforced phenolic foam (FPF) were characterized by Fourier transform infrared spectroscopy (FT-IR), scanning electron microscopy (SEM), pore morphology analysis, mechanical properties, thermal conductivity testing, and flame retardant performance analysis (LOI).

## 2. Materials and Methods

### 2.1. Materials

Poplar fiber was provided by Anhui Southeast Wood Industry Co., Ltd. (Wuhu, China). The fibers were pulverized, and the treated fibers were respectively screened with 40 mesh and 60 mesh screens. Fibers between 40 and 60 mesh were selected.

Phenol, polyformaldehyde, formaldehyde (37%), sodium hydroxide, concentrated hydrochloric acid (36%), phosphoric acid (>85%), p-toluene sulfonic acid, petroleum ether, 95% ethanol, acetic acid, glacial acetic acid, concentrated sulfuric acid, acetic anhydride, and other chemicals were supplied by Beijing Chemical Reagents Co., Ltd. (Beijing, China). 

### 2.2. Preparation of Phenolic Resin (PF) Foam

Phenol and formaldehyde solutions were added into a flask and stirred uniformly, and the temperature was increased to 85 °C. The first portion of polyformaldehyde was then added to the flask. The first portion of the NaOH solution was added to maintain the reaction temperature at 85 °C for 40 min. The second portion of polyformaldehyde and the first portion of the formaldehyde solution (37% concentration) were added to react with the NaOH solution at 82 °C for 40 min. The solution was then cooled down to 80 °C. The second portion of the formaldehyde solution and the third portion of the NaOH solution were then added, and the reaction time was 40 min at 80 °C. Finally, the viscous, brown-red phenolic resin foam with a high solid content was obtained by cooling the composite below 40 °C.

### 2.3. Acetylation Treatment of Poplar Fiber

The fibers were pulverized, and the treated fibers were respectively screened with 40 mesh and 60 mesh screens. Fibers between 40 and 60 mesh were selected.

The dry poplar fibers were first added into the NaOH aqueous solution at a mass fraction of 4%. The ratio of poplar fibers to NaOH aqueous solution was 1:4, and they were stirred for 2 h at 30 °C. The treated fibers were then washed to neutrality with deionized water, filtered and air-dried at 60 °C for 24 h, and finally packed and sealed for preservation.

Glacial acetic acid and acetic anhydride were added to the flask at a ratio of 3:4, and concentrated sulfuric acid was then added as a catalyst. The quality of the flask was 10%. The reaction time was 6 h under the condition of a constant temperature water bath at 50 °C. After the reaction, the sample was taken out, and the fibers were washed repeatedly with deionized water to neutrality. The fibers were then dried at 60 °C in a constant temperature oven for 24 h.

### 2.4. Preparation of Phenolic Foam Composites Reinforced by Plant Fibers

The numbers and formulations of all foam composites are listed in Table 1. Among them, No. 0 is the pure phenolic resin foam composite; No. 1 to 4 are the modified phenolic resin foam composites with 2%, 5%, 7%, and 10% addition of untreated poplar fiber; No. 5 to 8 are the modified phenolic resin foam composites with 2%, 5%, 7%, and 10% addition of acetyl-treated poplar fiber.

All foam samples were prepared using the one-shot and free-rise method. The chemical compositions of the samples are listed in Table 1. The PF and surfactant Tween-80 were first added into a 250 mL plastic beaker at room temperature and then stirred at 750 rpm for 3 min. Tween-80 is a commonly used nonionic surfactant in the preparation of foam materials. It can act as a stabilizer and auxiliary foaming agent in the production of foam materials. A blowing agent (petroleum ether) was then added to the mixtures and stirred at 750 rpm for 3 min. After the complex acid curing agent (p-toluenesulfonic acid:phosphoric acid:hydrochloric acid:water = 2:1:2:1) was added, the mixtures were stirred at 1000 rpm for 5 min. Each resulting viscous mixture was immediately poured completely into a foaming mold with a lid and then cured at 75 °C for 2 h. The fibers weight fractions incorporated into the material were 2, 5, 7, and 10 wt% based on the mass of the resin. PF and fibers were first added into a 250 mL plastic beaker at room temperature and then stirred at 750 rpm for 15 min. The remaining steps of the preparation were the same as those conducted to prepare pure PF. 

### 2.5. Characterization of Phenolic Foam Composites

Fourier transform infrared spectrometry (FTIR) testing was conducted, and the fibers and foam composites were characterized by the ATR method. About 5–10 mg finely ground powder for each sample was analyzed directly with a Nicolet 6700 spectrometer (Thermo Scientific, Pittsburgh, PA, USA) by the attenuated total reflectance method in the optical range of 500–4000 cm^−1^ and with 32 scans on average at a resolution of 4 cm^−1^.

Scanning electron microscopy (SEM) testing was carried out with a Hitachi S-4800 (Tokyo, Japan) to observe the appearances of the fibers and foam composites.

The apparent densities of all foam composites were measured according to the ASTM D1622-03 standard. Samples with dimensions of 30 × 30 × 30 mm^3^ were intercepted from each foam composite.

The compressive strength and modulus along the free upward direction of the foam composites measured by a universal testing machine (Instron 6022) in accordance with ASTM D1621-10. Samples with dimensions of 30 × 30 × 30 mm^3^ were intercepted from each foam composite, and the downward compressive speed of each sample was 2.5 mm/min. When the strain is between 0% and 10%, the maximum value of the stress–strain curve is the compressive strength, and the slope of the elastic deformation line on the curve is the compressive modulus. Five samples were tested for each foam composite, and the average value was obtained.

Pulverization ratio testing was also conducted. Samples with dimensions of 50 × 50 × 20 mm^3^ were intercepted from each foam composite, and were then weighed (M1). The 50 × 50 mm side of each sample was then placed on sandpaper, and a 200 g weight was placed on top of the foam composite. The sample was then pulled back and forth on the sandpaper 30 times with a pull distance of 250 mm each time. The remaining mass of the sample was then weighed as M2. The pulverization rate was determined according to the formula (*M*1 *− M*2)/*M*1 × 100%. Four samples were tested for each foam composite, and the average value was obtained. 

The thermal conductivity of the foam composites was measured by a Hot Disk TPS 2500 thermal conductivity analyzer. Samples with dimensions of 50 × 50 × 15 mm^3^ were intercepted from each foam composite. Three samples were tested for each foam composite, and the average value was obtained.

The limiting oxygen index (LOI) of each foam composite was determined by an HC-2 type instrument (ASTM), and tested according to ASTM D2863. Samples with dimensions of 100 × 10 × 10 mm^3^ were intercepted from each foam composite, and the results were averaged.

Determination of foam acidity: Took phenol formaldehyde foam 5 g, grinded it, and immersed it in 100 mL distilled water for about 30 min. Because phenolic foam powder is insoluble in water, we filtered out the insoluble matter, determined the pH value of the aqueous solution, and judged the pH value of the foam acid according to the pH value of the solution.

Foam hydrophobicity test: Samples with dimensions of 40 × 40 × 10 mm^3^ were intercepted from each foam composite, samples were immersed in red ink, and samples were taken at 24 h and 72 h.

## 3. Results and Discussion

### 3.1. Characterization of Poplar Fiber, Poplar Fiber-NaOH, and Poplar Fiber-NaOH Acetylation 

To solve the compatibility problem between poplar fibers and resin, compatibilizing treatment is required. The wood fiber was washed with 10% NaOH solution and then acetylated with an acetylation reagent. The results are presented in Figure 1.

Figure 1a presents the electron micrographs of untreated poplar fiber (1), sodium hydroxide solution-pretreated poplar fiber (2), and acetylated poplar fiber (3), from which it can be seen that after sodium hydroxide solution treatment, some hemicellulose and pectin on the surface of the fiber were removed, leaving a single fiber with a neat appearance. The single fiber was shrunken due to the swelling effect of the sodium hydroxide solution, which increased the contact area between the fiber and the resin substrate, and the fiber could therefore be effectively dispersed in the resin substrate. After acetylation, the surface of the fiber was successfully coated with an ester substance. The esterified layer can reduce the polarity of the fiber, resulting in the better dispersion of the fiber in the resin, and fiber agglomeration can be effectively avoided. Figure 1b presents the infrared spectra of the fibers, in which the intensity of the -OH absorption peak at 3419 cm^−1^ is weaker than that of the pretreated fiber, indicating that the acetylating agent reacted with the hydroxyl group in the fiber, thus reducing the hydroxyl groups in the product after acetylation modification. Additionally, 1743 cm^−1^ is the absorption peak of C=O. The peak disappeared after sodium hydroxide treatment, indicating that hemicellulose and lignin were removed by sodium hydroxide. A significant increase in the peak value occurred after acetylation, which indicates that the acetylation was successful, and the absorption peaks at 1374 cm^−1^ (C-H) and 1241 cm^−1^ (-CO-) in the acetyl group also indicate that the acetyl group was introduced into the modified product. Figure 1c is a representation of the thermal stability of the fiber, from which it can be seen that the modified fiber presented no major changes. Before and after modification, the fiber had only one degradation stage, but the acetylated fiber was thermally stable. The thermal stability was greatly improved, as was the residual carbon ratio, which indicates the removal of hemicellulose with poor thermal stability from the fiber surface. This change improved the thermal stability of the fiber.

The acetylated fiber presented better dispersibility, reduced agglomeration, and improved compatibility with the resin substrate. The thermal stability was also improved, which is crucial for the preparation of the foam.

### 3.2. Characterization of PF, FPFs, and FTPFs

#### 3.2.1. Fourier Transform Infrared (FT-IR) Spectroscopy of Foam

FT-IR spectroscopy was employed to determine whether the fiber established a stable chemical linkage in the phenolic resin foam. Figure 2 presents the chemical structures of PF, FPF, and FTPF. A typical hydroxyl absorption peak is present at 3200–3400 cm^−1^, 2850–2930 cm^−1^ is the position of the C-H absorption peak, and 1450–1600 cm^−1^ is the absorption peak of the aromatic ring in the phenolic resin. As shown in the figure, the spectra of FPF and FTPF are similar to that of the conventional PF, which indicates that the chemical structure of phenolic foam after fiber modification was similar to that of traditional phenolic foam. During the preparation process of fiber-modified phenolic foam, no chemical cross-linking between the fiber and phenolic foam occurred, and the fibers were only embedded in the resin as a filler (as shown in the optical micrograph embedded in the infrared spectra). 

#### 3.2.2. Apparent Density and Cell Density of Foams

The apparent density is a key indicator for the evaluation of the performance of phenolic foam. The volume of the phenolic resin was about 10–20% of the volume of the foam. The apparent density of the foam includes the resin (cell wall) density and the cell density, and thus the partial properties of the phenolic foam, can be expressed in terms of apparent density. Table 2 lists the apparent density, cell median diameter, and cell density values of the PF, FPFs, and FTPFs. Because the weight of the fiber was much heavier than the weight of the foam, an increase in the apparent density of the fiber-modified phenolic foam was inevitable. Moreover, as the fiber content increased, the apparent density of the foam also increased. It can be seen from the table that the densities of the foam materials presented an increasing trend with the increase of the fiber addition amount. However, the addition of fibers affected the cells of the foam, which resulted in the significant variation of the cell density. The cell density of the reference PF was 4.47 × 10^5^ cells/cm^3^. With the 2% addition of fiber, regardless of whether the fiber was acetylated or not, the density of the cells increased. This can be attributed to the performance of the softwood powder as a nucleating agent, which promoted the formation of cells; this is consistent with previous reports [30]. With the increase of the addition amount of unmodified fiber, the cell density gradually decreased; this is because the fiber content was too high and agglomeration occurred among fibers, causing the morphology of the cells to be broken and resulting in a decrease in cell density. However, in the FTPFs, although the cell density decreased with the increase of fiber content, the degree of reduction was less than that in the FPFs; this was due to the weakening of fiber agglomeration after acetylation modification, and the fibers had better compatibility with the resin substrate. The cell density of the FTPF-5% foam was the highest, reaching 5.24 × 10^5^ cells/cm^3^ and exceeding that of PF. The foams with higher cell density tended to display better performance.

#### 3.2.3. Morphologies of Foams

The most important factor of foam is the morphology of its cells. The performance of the thermal insulation material has a significant relationship with the closed cell ratio of the material. The morphologies of the PF, FPFs, and FTPFs foam were observed by scanning electron microscopy. The SEM images and cell distributions of the different types of foam are presented in Figure 3.

Figure 3 reveals that most of the cells in the foam were closed, which is the most basic condition for thermal insulation. However, some perforations can be observed in the photograph, and may have been caused by the evaporation of water in the foam resins. Partial cracking or debris was also observed in all foams due to cutting problems during sample preparation. PF exhibited a regular and uniform cell structure with an average cell size of 103.31 μm and a cell size distribution of 40–160 μm. With the addition of a small amount of fiber, the average diameter of the foam cells was reduced. When fiber was added in an amount exceeding 5%, the pore size distribution of the foam was wider and the average diameter was increased, which is primarily because the fibers did not mix well in the resin substrate. Additionally, the fiber itself easily agglomerated. To solve this phenomenon, the fiber was acetylated. It was found that the FTPF foams exhibited a relatively regular and uniform cell structure compared to the FPF foams. The foam cells of FTPF-5% exhibited the most perfect structure, their average diameter (96.67 μm) was lower than that of the PF foam, and the cell distribution was mainly 80–100 μm. The smaller and more uniform cell morphology can lead to excellent performance.

#### 3.2.4. Mechanical Properties of Foams

The compressive property is one of the indexes by which to evaluate the mechanical properties of materials; therefore, the compressive strength and modulus of the modified phenolic foams were evaluated. The compression of phenolic foam generally occurs in three main stages. In the early stage of compression, the strength of the cell wall and the internal gas pressure maintain the basic shape of the cell. Cell wall breakage and collapse and compaction then occur, which makes the cell deformation of stress in the sample dispersion effect the main factors influencing the compressive strength of size [31]. Figure 4a shows the compressive strengths of the PF, FPFs, and FTPFs. When the fiber content in the FPFs was low, the compressive performance of the foam was increased. The main reasons for this are the nucleation of the fiber, which resulted in the growth of foam on the fiber, and the relatively uniform and compact structure of the cell. With the increase of the fiber content, the compressive strength of the foam decreased gradually. This is primarily due to the increase of fiber addition, which led to fiber agglomeration and lower compatibility between the fiber and resin substrate. During the foaming process, the agglomerated fibers broke up the uniformity of the original resin foam, resulting in large, uneven foam cells and even perforation, which directly led to the reduction of the mechanical properties of the foam materials (Figure 5). When the fiber content was increased to 10%, the strength of the foam was the lowest (0.07 MPa); low-strength foam materials lose practical use value. To address this, the fiber was modified by acetylation to obtain FTPFs. The strength was significantly increased; FTPF-5% presented the highest strength (0.18 MPa), an increase of 28.5% compared to the strength of the PF foam (0.14 MPa). The main reasons for this are that, after the acetylation of the fiber, polarity was reduced and the compatibility between the fiber and resin was enhanced, which reduced fiber agglomeration. During the foam formation stage, fiber played a role as the nucleating agent. The cells were uniform and small, and no large cells were formed, thereby increasing the strength of the foam material. Figure 4b shows that the compressive modulus was approximately the same as the compressive strength structure. The modulus of the modified poplar was slightly higher than that of the unmodified poplar, indicating that the modification of fiber can improve the interface compatibility between the fiber and foam. The best compressive performance was exhibited by FTPFs-5%; its compressive modulus was 8.05 MPa, presenting an increase of 37.9% compared with PF (5.83 MPa).

The inserted picture shows the state of wood fibers mixed with resin.

#### 3.2.5. Pulverization Ratios of Foams

The high pulverization ratio of phenolic foam is restricted by its application range, so it is necessary to test and analyze the pulverization ratio of modified phenolic foam. Figure 6a presents the pulverization ratios of the PF, FPFs, and FTPFs. The pulverization ratio of pure phenolic foam (PF) was the highest at 9.6%, indicating brittleness. The reason for this may be that the phenolic foam itself has a short molecular chain, and it can easily drop powder and slag when it is subjected to force and friction. With the increase of fiber addition, the foam pulverization changed, and a small amount of fiber led to the decrement of the pulverization ratio of the foam. The main reason for this is that fiber helped to form small cells, and the cell walls were supported by the protection of the surrounding fiber. Fiber can withstand a certain friction, improve the wear resistance of phenolic foam, and thus reduce the loss of the quality of the foam. However, with the increase of fiber addition (>5%), the agglomeration of fibers led to the bursting or perforation of the foams, which destroyed their uniform structure. The burst foam structure was more likely to fall off, thus increasing the pulverization rate of the foam material. This is not conducive to the actual utilization of the foam material. When the fiber was acetylated to prepare FTPFs, the fiber mixed well with the resin, which greatly reduced the probability of the crushing and perforation of the foam. Fibers exhibit a foam molding role, supporting role, and fiber itself wear resistance, and the distribution of fibers in the resin can result in diverse forces of foams (Figure 6c). The relatively short and rigid structure of the original foam (PF) could not sufficiently diffuse stress in the foams, and the phenomenon of stress concentration therefore easily occurred. However, in the FTPFs, the fiber greatly dispersed stress and thereby improved the mechanical stability of the foam. FPF-5% and FTPF-5% had lower pulverization ratios (7.4% and 6.5%, respectively), presenting decrements of 22.9% and 32.3% compared to PF (9.6%). Via a comparison with modified foams reported in the existing literature, particularly those modified with biomass materials, it was found that the fiber-modified phenolic foam presented in the current work had a better pulverization rate (Figure 6b) and exhibited satisfactory results.

#### 3.2.6. Thermal Conductivity and Limiting Oxygen Index (LOI) of Foams

The thermal conductivity of an object is an important index by which to evaluate its thermal insulation performance. Porous plastic foam structures contain large volumes of gas, so gas conductivity is often a key factor in heat transfer. However, the heat transfer of foam is mainly conducted through solid polymers, which means that the thermal conductivity of foam is affected by both solid and gas. Besides the factors that affect the thermal conductivity of phenolic foam, such as the density of the resin, the size distribution of foams, the median diameter of foams, and the thermal conductivity of solid polymers, added fillers or modifiers also affect the thermal conductivity of the foam to some extent [34]. Generally, good thermal insulation materials should have small, uniform, and closed cells. Figure 7a presents the thermal conductivities of the PF, FPFs, and FTPFs. The thermal conductivity of PF was 0.058 W/mK. The thermal conductivities of the FPFs increased with the increase of fiber addition. A small amount of fiber addition can ensure the stability of the thermal conductivity; the thermal conductivity of FPF-5% was slightly lower than that of PF. The main reason for this that the foam material had better cell uniformity, and the size of the foam cell was smaller than that of PF. When the addition amount of fiber was further increased, the cell wall breaking and perforation caused by fiber agglomeration led to a significant increase in the thermal conductivity of the foam materials. In the acetylated modified foam (FTPFs), the fiber and resin substrates were well integrated, and the probability of large perforation was reduced. Therefore, the thermal conductivity of the foams could be well controlled. FTPF-5% foam exhibited the lowest thermal conductivity of 0.056 W/mK.

Building fires occur frequently, and therefore building insulation materials must be flame retardant. As a widely used building insulation material, the flame retardancy of phenolic foam has always been an important factor that affects its application. Therefore, it is necessary to analyze the flame retardancy of the modified foams. The limiting oxygen index (LOI) is the minimum oxygen concentration required for the combustion of materials in a mixture of oxygen and nitrogen.

The LOI values of the PF, FPFs, and FTPFs are presented in Figure 7b. The LOI values of the FPFs presented a significant reduction compared to that of PF; the decreasing trend of the LOI value became more obvious with the increase of the fiber addition amount. The reason for this phenomenon is the flammability of fiber itself. After acetylation, the LOI values of the FTPFs were higher than those of the FPFs. This is because, after acetylation processing, hemicellulose on the surface of the fiber, which can easily decompose, was removed. Additionally, the fiber and resin substrate could more easily fuse, thereby reducing the generation of foam debris. The LOI of FTPF-5% was 33.9%, slightly higher than that of the PF foam (33.3%), which was mainly related to the cell densities of the foams. The cell density of the FTPF-5% foam was higher; the higher proportion of carbon atoms in the same plane resulted in easier carbonization of the foam, which improved its self-extinguishing performance. The sample with the lowest LOI value was FPF-10% (27.7%); it was therefore a refractory material (LOI > 26%), indicating that fiber-modified phenolic resin could be used as a building insulation material.

Curing agent is an indispensable part of in the process of preparing foam and inorganic acid; because of its fast curing speed and low price, it is widely used, but it can cause equipment corrosion in the curing process and the residual acid in the foam can cause corrosion of metal materials in building walls. In order to solve this problem, this paper adopted complex acid curing phenolic foam. By measuring the acidity of the two foams (Figure 8a), it was found that the acidity of complex acid curing foam was decreased obviously. The foam material itself has hydrophobic properties (Figure 8b), and the hydrophobic angle reaches 101°. After immersion for 24 h and 72 h, it was found that the permeability of the aqueous solution is limited and only permeates the surface of the foam material and cannot enter the interior of the foam. Therefore, the acidity of the foam solution is greatly reduced, which can effectively protect the steel beneath the foam.

## 4. Conclusions

In this article, via the characterization of the morphology, mechanical properties, crushing ratio, and thermal flame retardant properties of phenolic resin foams, it was found that the prepared modified fiber foams (FTPFs) were better than unmodified fiber foams (FPFs). The most outstanding performance of the FTPFs was the foam with an acetylated fiber addition amount of 5% (FTPF-5%). Compared to PF, the cell distribution of FTPF-5% was narrower, its cell diameter was reduced, its cell density was increased, and its compressive strength and modulus were increased by 28.5% and 37.9%, respectively. Additionally, the pulverization ratio was successfully reduced from 9.6% to 6.5%, and the thermal insulation performance and flame retardant performance (LOI) were slightly improved. The main reasons for this are that, after acetylation treatment, the polarity of poplar fiber decreased, the phenomenon of fiber agglomeration lessened, and the compatibility between the fiber and phenolic matrix improved. Additionally, the addition of fiber increased the viscosity of the resin, and was conducive to the nucleation of the foam, thereby reducing the cell size and increasing the cell density of the material. The porous structure of foam directly improved the mechanical properties of the material, and simultaneously ensured the insulation and flame retardant properties of the foam material.

## Figures and Tables

**Figure 1 materials-13-00148-f001:**
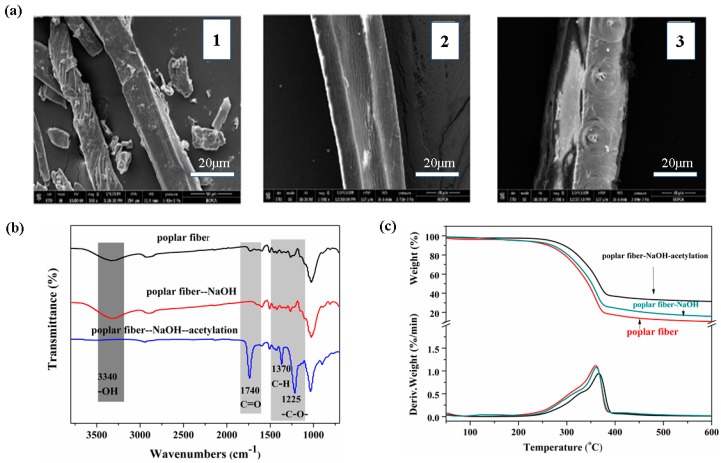
(**a**) SEM figures of poplar fibers: (1) Poplar fiber, (2) poplar fiber-NaOH, and (3) poplar fiber-NaOH-acetylation. (**b**) FTIR spectra of poplar fibers. (**c**) TGA and DTG curves of poplar fibers.

**Figure 2 materials-13-00148-f002:**
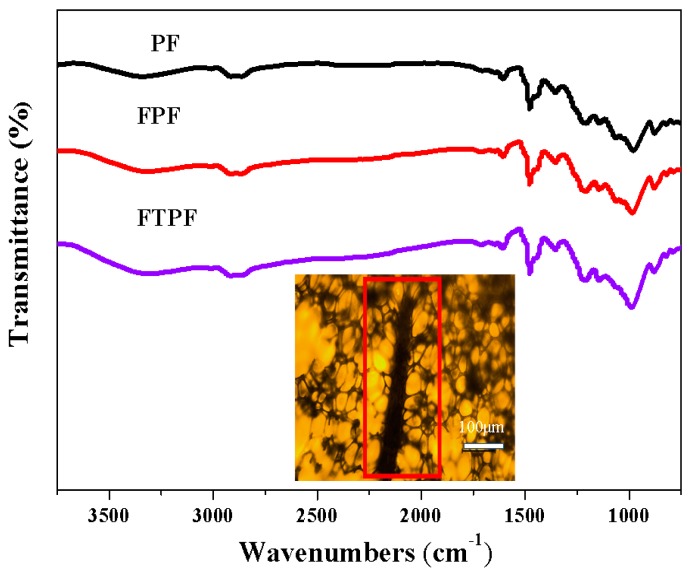
FTIR spectra of phenolic foam (PF), FPF, and wood fiber-reinforced FPF (FTPF).

**Figure 3 materials-13-00148-f003:**
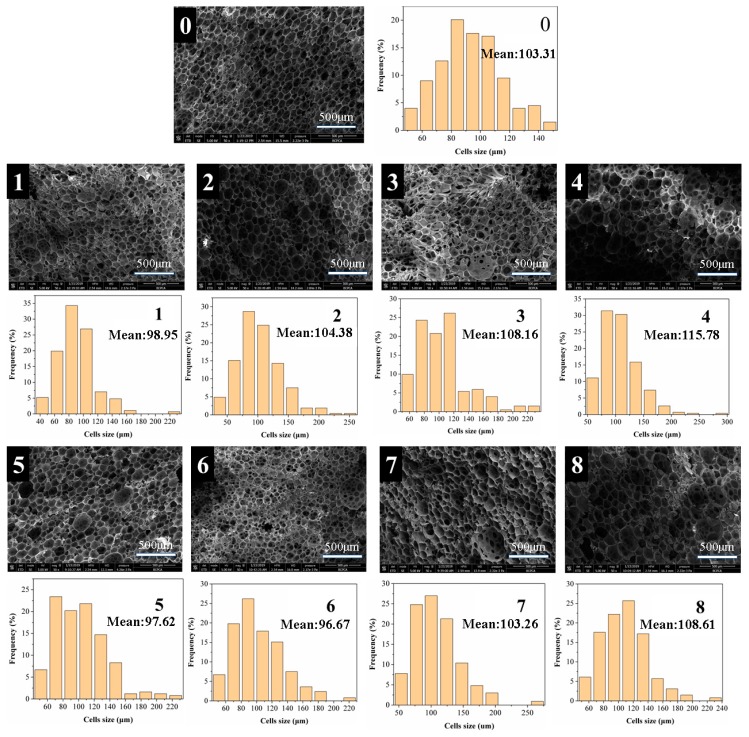
SEM images and cell size distributions of the PF (**0**), FPFs (**1**–**4**), and FTPFs (**5**–**8**).

**Figure 4 materials-13-00148-f004:**
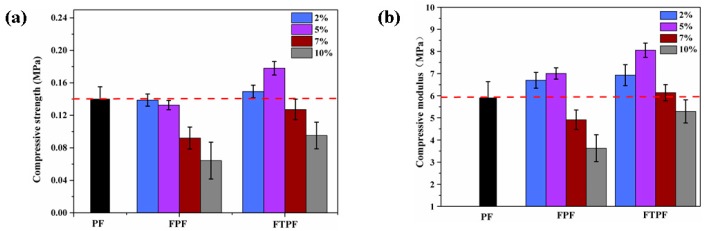
Normalized (**a**) compressive strength and (**b**) compressive modulus of PF, FPFs, and FTPFs.

**Figure 5 materials-13-00148-f005:**
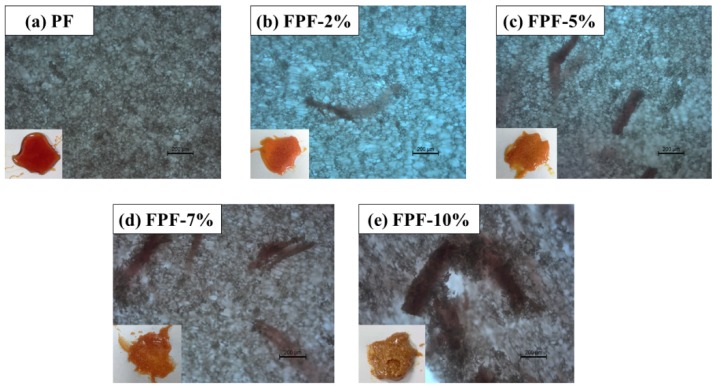
The distribution of fibers in foams.

**Figure 6 materials-13-00148-f006:**
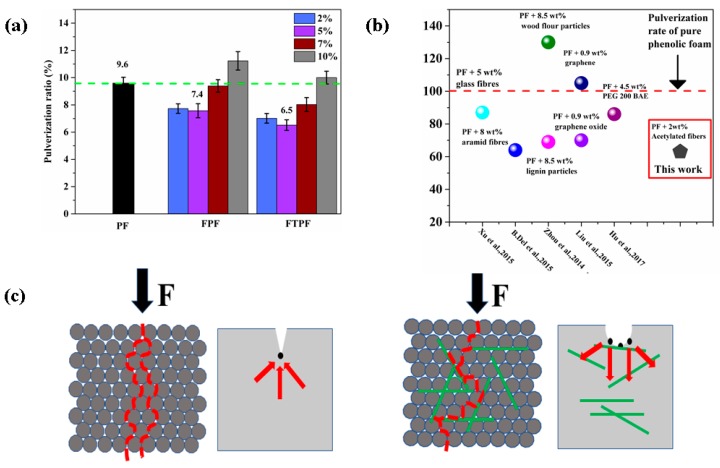
(**a**) Pulverization ratios of the PF, FPFs, and FTPFs. (**b**) Comparison between toughened modified foam resins in the existing literature and the preferred foam resins in the present study [4,12,31,32,33]. (**c**) Model diagrams of PF and FPF under loading.

**Figure 7 materials-13-00148-f007:**
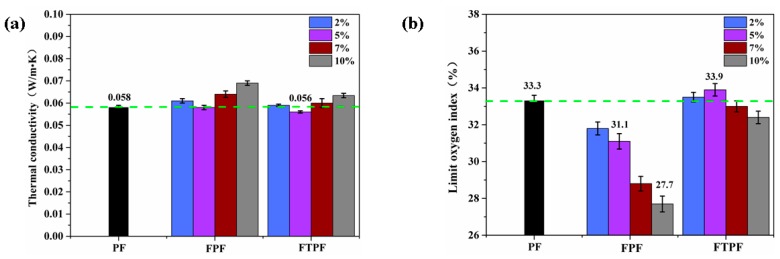
(**a**) Thermal conductivity and (**b**) limiting oxygen index of the PF, FPFs, and FTPFs.

**Figure 8 materials-13-00148-f008:**
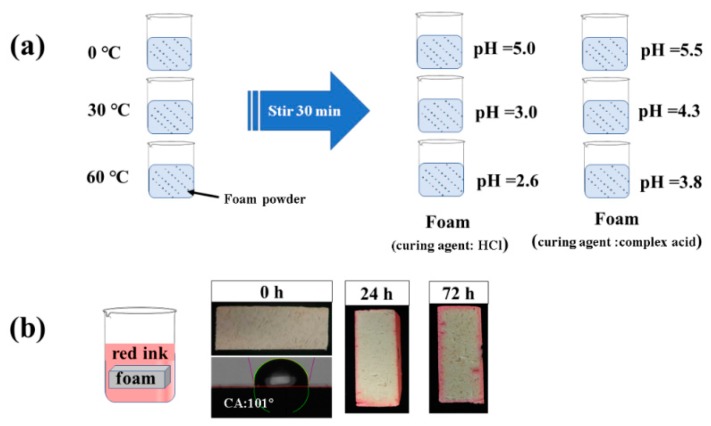
(**a**) Determination of foam acidity and (**b**) foam hydrophobicity test of FTPF.

**Table 1 materials-13-00148-t001:** Formulations of phenolic foam composites.

Foam Number	Foam Name	Tween-80 (wt%)	Petroleum Ether (wt%)	Complex Acid Curing Agent (wt%)	Original Poplar Fiber (wt%)	Acetylated Treated Fiber (wt%)
0	PF	10	6	15	/	/
1	FPF-2%	10	6	15	2	/
2	FPF-5%	10	6	15	5	/
3	FPF-7%	10	6	15	7	/
4	FPF-10%	10	6	15	10	/
5	FTPF-2%	10	6	15	/	2
6	FTPF-5%	10	6	15	/	5
7	FTPF-7%	10	6	15	/	7
8	FTPF-10%	10	6	15	/	10

**Table 2 materials-13-00148-t002:** Apparent density, cell median diameter, cell density, and porosity of foams (0–8).

Foam Number	Foam Name	Apparent Density (kg/m^3^)	Cell Median Diameter (μm)	Cell Density *N_F_* (10^5^ Cells/cm^3^)	Porosity *f* (%)
0	PF	46.31	103.31	4.47	94.27
1	FPF-2%	48.74	98.95	4.76	93.79
2	FPF-5%	51.83	104.38	4.35	93.07
3	FPF-7%	53.44	108.16	3.06	92.59
4	FPF-10%	58.42	115.78	2.70	91.87
5	FTPF-2%	48.38	97.62	4.94	93.79
6	FTPF-5%	50.97	96.67	5.24	93.07
7	FTPF-7%	52.52	103.26	3.95	92.59
8	FTPF-10%	57.21	108.61	3.40	91.87

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
