# Peer review of "Phenolic Resin Foam Composites Reinforced by Acetylated Poplar Fiber with High Mechanical Properties, Low Pulverization Ratio, and Good Thermal Insulation and Flame Retardant Performance"

_materials, 2019, doi:10.3390/ma13010148_

Round 1

Reviewer 1 Report

The paper is based on a topic that, in principle, could be interesting.

Nevertheless, the text in the manuscript is difficult to read in some parts and to understand in others. A careful check of the sentences and a proofreading are needed.

In regard to the scientific content, the paper appears as a technical report, with comments not very much pertinent. In the following my main concerns:

Some crucial information are lacking: foam density and porosity. The mechanical response of a foam is dependent on both, and on the reinforcement content, other than quality of the dispersion.

The dispersion of fibres was neither qualitatively shown or quantified. How can the authors make any statement without knowing how much fibres are dispersed and how much are aggregated?

They could try to use a mathematical model for composites to estimate the (expected) mechanical response of the foam and then discuss the differences.

Is’s true that cell size and cell size distribution have a role in the mechanical response, but it is not a major one. The increase in modulus and strength are mainly related to the density and the reinforcement content (for 2% and 5% systems), while cell opening can be the reason for the decrease at higher percents. The authors cannot neglect the foam density when they compare the mechanical response of different foams. For example, the increase in modulus (in general limited) and in strength (in general very limited) can be due to the higher density of reinforced foams with respect to the neat foam. The normalized compressive modulus and strength must be calculated (please refer to the reference book in foam’s mechanics “Cellular Solids” from Gibson and Ashby to know ho to handle this task) and then compared.

The terms used by the authors are not clear: what is the cell wall density? In foams I never saw such term to refer to foams.

The morphology of the foams is not clear from the picture they showed in Fig. 3, hence it can be checked and several statements of the authors cannot be checked.

The performance coming from the addition of the fibres are not clear. It seems that there isn’t a univocal indication of the performance gain. In same cased the 2% filled sample is better, in other the 5%.

Minor issues

Figure numbering must be checked

Add a picture that shows the actual fibre dispersion quality in the matrix

Reviewer 2 Report

The presented article by Liu et al., depicts preparation and characterization of phenolic foams with the addition of both modified and unmodified fiber in different contents.

The authors described modification technique and conducted material characteristics. In general the paper is of an average value with respect to broadening knowledge and extending methods of plant fiber reinforced composites. The way the results are presented is clear and legible. Figures and tables understandable. The description of the results forms a unified whole creating a logical cause and effect sequence.

I recommend the paper to be publish in the Materials journal after making minor revision.

Below are some comments with a request to explain and respond to them:

The authors should explain to the reader what type of fiber we are dealing with. I think that term 'popular fiber' is not sufficient. On what basis did the authors apply such modification conditions, both alkalization and acetylation?

When preparing phenolic foams, it is very important to carefully and precisely control the time of adding individual ingredients. I am asking for a more detailed description of the method of preparation of composites (including times for each steps, component addition) in section 2.4. What does it mean: Tween-80? On which apparatus FTIR analysis was performed?

Despite all advantages mentioned in the article, phenolic foams however contains an acid catalyst. Upon exposure to water such as rain, the acid catalyst may be extracted from the phenolic foam. This could cause a problem when metallic materials are in contact with the phenolic foam. For example, phenolic foam used as roof insulation could corrode steel decks beneath the foam. I suggest to perform and add the results of foams pH measurement (both cold and warm methods).
